# The Detection of Extensively Drug-Resistant *Proteus mirabilis* Strains Harboring Both VIM-4 and VIM-75 Metallo-β-Lactamases from Patients in Germany

**DOI:** 10.3390/microorganisms13020266

**Published:** 2025-01-25

**Authors:** Moritz Fritzenwanker, Jane Falgenhauer, Torsten Hain, Can Imirzalioglu, Trinad Chakraborty, Yancheng Yao

**Affiliations:** 1Institute of Medical Microbiology, Justus Liebig University Giessen, Schubertstrasse 81, 35392 Giessen, Germany; moritz.fritzenwanker@mikrobio.med.uni-giessen.de (M.F.); jane.c.falgenhauer@mikrobio.med.uni-giessen.de (J.F.); torsten.hain@mikrobio.med.uni-giessen.de (T.H.); can.imirzalioglu@mikrobio.med.uni-giessen.de (C.I.); trinad.chakraborty@mikrobio.med.uni-giessen.de (T.C.); 2German Center for Infection Research (DZIF), Partner Site Giessen-Marburg-Langen, Justus-Liebig University Giessen, 35392 Giessen, Germany; 3Institute for Hygiene and Environmental Medicine, Justus Liebig University Giessen, Schubertstrasse 81, 35392 Giessen, Germany

**Keywords:** *Proteus mirabilis*, carbapenemases, VIM-75 and VIM-4, multi-drug-resistance, plasmid

## Abstract

*Proteus mirabilis* is a well-known opportunistic pathogen predominantly associated with urinary tract infections. It exhibits natural resistance to multiple antibiotics, including last-resort options like colistin. The emergence and spread of multidrug-resistant *P. mirabilis* isolates, including those producing ESBLs, AmpC cephalosporinases, and carbapenemases, are now more frequently reported. The most common carbapenemase types found in *P. mirabilis* are KPC-2, IMP, VIM, NDM, and OXA-48. We sequenced the genomes of three carbapenem-resistant *P. mirabilis* isolates harboring both *bla*_VIM-4_ and *bla*_VIM-75_ from Germany using both short-read and long-read sequencing techniques. We found that the isolates were only distantly related genetically. Both *bla*_VIM-4_ and *bla*_VIM-75_ genes were located on a class I integron, which in two cases was located on the chromosome and in one case on a plasmid. This is the first report on the complete genomes of *P. mirabilis* strains harboring a rare genetic element encoding both *bla*_VIM-4_ and *bla*_VIM-75_. Our results emphasize a key role for class 1 integrons in the transmission of VIM carbapenemases in *P. mirabilis*.

## 1. Introduction

*Proteus mirabilis* is a commensal of the human digestive tract, also present in various environmental sources, particularly in sewage water and soil. In the medical context, the species is mostly known for causing urinary tract infections and infections originating from the intestines, such as peritonitis. When these infections breach the immune system or overcome medical interventions, *P. mirabilis* can cause bloodstream infections (BSIs) and subsequently septicemias. Typical pathogenicity factors include fimbriae, adhesion molecules, hemolysis (not visible on Columbia sheep blood agar), and flagella, which enable strains to swarm over agar plates in a typical specific pattern [1,2,3,4]. Common treatment options include aminopenicillins, second- and third-generation cephalosporines, trimethoprime–sulfamethoxazole, and fluoroquinolones. Like other *Enterobacterales*, *P. mirabilis* can develop or acquire resistance to any or all of these substances, demanding physicians to draw and reserve antibiotics, most typically on carbapenems. Risk factors are linked with the acquisition of multidrug-resistant (to three or more classes of antimicrobials) strains.

*Proteus* are members of the family *Morganellaceae* and have a certain degree of intrinsic resistance to imipenem. However, this intrinsic resistance can usually be overcome at high doses or in infections at sites with high concentrations of imipenem, especially in the urinary tract. Nevertheless, *Proteus* can also acquire carbapenemases, which render them virtually immune to treatment with carbapenems. In Germany, *P. mirabilis* isolates harboring carbapenemases are rare, but pose a significant threat to human health when encountered in potentially life-threatening infections. *Proteus* are also naturally resistant to polymyxins like colistin, which is generally only rarely used because of severe neurological and nephrological side effects but remains as one of the important last-resort antibiotics for infection with carbapenem-resistant Gram-negative bacteria [5].

Recently, the emergence and spread of carbapenemase-producing *P. mirabilis* encoding types *bla*_KPC-2_, *bla*_NDM_, *bla*_OXA-23_, and *bla*_OXA-48_ have been reported in many countries around the world [6]. The acquisition of *bla*_VIM_ genes has occurred in Greece, Italy, and Bulgaria [7,8]. In Germany, carbapenemase-producing *Enterobacterales* are increasing in prevalence [9]. Nationwide surveillance reports note that *E. coli* and *Klebsiella* species are the most common carbapenemase-producers, while *P. mirabilis* harboring carbapenemases have only rarely been detected.

We carried out a surveillance study of carbapenem-resistant Gram-negative bacteria in the state of Hesse in Germany, in which carbapenem-resistant *Enterobacterales* collected within a three-year period were subjected to whole-genome sequencing. In the present study, we included three *P. mirabilis* isolates exhibiting phenotypic carbapenem resistance and containing both *bla*_VIM-75_ and *bla*_VIM-4_ genes encoding VIM carbapenemase. Herein, we present a detailed description of the nature of the resistance elements in the context of their genomes and their plasmids.

## 2. Materials and Methods

### 2.1. Study Design and Bacterial Isolates

An epidemiologic surveillance study on carbapenem-resistant Gram-negative bacteria was conducted from 2017 to 2019 across 61 hospitals in the State of Hesse in Germany. This molecular epidemiologic investigation yielded 520 *Enterobacteriaceae* isolates that exhibited non-susceptibility to at least one carbapenem, including five *P. mirabilis* from various hospitals in 2018 and 2019. Three isolates with duplicated VIM alleles were identified from the genome analysis with the short-read sequence data and no known carbapenemase genes were found in two other isolates. Long-read sequencing was then performed for the three *bla*_VIM_-carrying isolates for future genomic analyses. They were isolated from groin swabs, blood cultures, and rectal swabs and derived from three epidemiologically unrelated patients aged 82, 80, and 92 years in three different hospitals, respectively, in Hesse, Germany, in 2019.

### 2.2. Antibiotic Susceptibility Testing

The antibiotic susceptibility test was performed using the VITEK^®^2 system (bioMérieux, Nürtingen, Germany) and interpreted in accordance with EUCAST guidelines. The quantitative MICs of imipenem and meropenem were determined using a Liofilchem MIC Test Strip (bestbion dx GmbH, Köln, Germany). Taxonomy was validated by utilizing MALDI-TOF-MS (Vitek MS, bioMérieux, Nürtingen, Germany).

According to the criteria for multidrug-resistant (MDR), extensively drug-resistant (XDR), and pandrug-resistant (PDR) bacteria by the ECDC and/or CDC [10], our three isolates analyzed belonged to XDR (extensively drug-resistant) because they were non-susceptible to at least 1 agent in all but 2 or fewer antimicrobial categories (see Table 1).

### 2.3. Whole-Genome Sequencing and Genome Analyses

Short-read whole-genome sequencing, post-sequencing quality control, and assembly were performed as described previously [11]. For detailed genomic characterization, these isolates were re-sequenced using PacBio long-read sequencing as described previously [12]. Genome assembly following quality control was performed using ASA^3^P [13]. Plasmid incompatibility (Inc) groups, antimicrobial resistance genes (ARGs), and insertion sequences (ISs) were identified using the Center for Genomic Epidemiology website (https://cge.food.dtu.dk/ accessed on 13 December 2023) [14,15] and ISFinder [16]. To identify the VIM alleles, blastN in the NCBI database screening was performed and validated with the bldb database (www.bldb.eu accessed on on 5 December 2024) [17]. oriTfinder (https://bioinfo-mml.sjtu.edu.cn/oriTfinder/ accessed on 5 December 2024) was used to determine virulent factor genes and predict OriT sites [18]. Mobile genetic elements (MGEs) associated with the ARCs were detected with the web tool MobileElementFinder [19]. The genetic structure surrounding the VIM genes was annotated and visualized by using Galileo AMR of ARC Bio [20]. To detect CRISPR-Cas sequences, CRISPRCasFinder was used [21].

### 2.4. Ethical Approval

Ethical approval was sought from the Ethics Committee of the State Medical Association of Hesse in Frankfurt/Main. The Committee decided on 24 January 2018 that ethical approval for the project was not necessary as this study’s patient data were rendered anonymous [11].

## 3. Results

### 3.1. Antibiotic Susceptibility

The isolates were phenotypically resistant to β-lactams (ampicillin, ampicillin–sulbactam, piperacillin, piperacillin–tazobactam, cefepime, cefpodoxime, cefotaxime, ceftazidime, cefuroxime, imipenem, and aztreonam), fluoroquinolones (ciprofloxacin, moxifloacine, and ofloxacin), aminoglycosides (gentamicin, tigecycline, and trimethroprime–sulfamethaoxal) and displayed reduced sensitivity towards to meropenem and ertapenem (Table 1). The colistin test indicated an MIC of >64 µg/mL for isolates Survcare401 and Survcare357.

### 3.2. Whole-Genome Sequencing and Genomic Features

The draft genomes of the isolates from the short-read sequencing exhibit a total length between 4.118 and 4.250 Mb. In combination with long-read-sequencing, the chromosome and one to three plasmids of each isolate were circular-closed (Table 2). The completed chromosomal sequence of Survcare401 is 4,218,249 bps in size and encodes 3852 CDSs, 22 rRNAs, and 86 tRNAs. The annotated CDSs are shown in Appendix A in detail. The genomic comparison of the three isolates revealed significant sequence orthologue similarity among them through BRIG (Appendix A).

### 3.3. Antimicrobial Resistance Genes and the Co-Occurrences of Both bla_VIM-75_ and bla_VIM-4_

The genomes each contained a large number of antimicrobial resistance genes (ARGs), which localized mostly on chromosomes (Table 2). Twenty-six different ARGs were identified on Survcare401, which belonged to seven antibiotic classes, including aminoglycoside (*armA*, *aac(6′)-IIc*, *strA*, *strB*, *aac(3)-IId*, and *aph(3′)-Ic*), beta-lactam (*bla*_VIM-4_, *bla*_VIM-75_, *bla*_CTX-M-15_, *bla*_TEM-1_, and *bla*_TEM-2_), marcrolide/lincosamide/streptogramin B (*lnu(F)*), (*mph(E)* and *msr(E)*), phenicols (*cat* and *catA1*), sulfonamides (*sul1* and *sul2*), tetracycline (*tet(J)*), and trimethoprim (*dfrA1* and *dfrA17*). A resistance gene against quaternary ammonium compounds (Δ*qac*E) was also identified.

The ARG repertoires of the isolates Survcare357 and Survcare372 were like Survcare401, differing only in the genetic location for some ARGs of Survcare372 which were located on a plasmid instead of the chromosome (Table 2).

Of note, both the *bla*_VIM-75_ and *bla*_VIM-4_ genes were embedded in a single class 1 integron whose genetic structure comprises the gene cassette [*bla*_VIM-75_, *aac(6)-IIc*, *bla*_VIM-4_, ∆*qacE*, and *sul1*] as identified on the chromosomes of Survcare401 and Survcare357 and on plasmid p372-3 of Survcare372, as shown in detail in Figure 1. The *bla*_VIM-4_ and *bla*_VIM-75_ genes are variants of *bla*_VIM-1_, each with a single amino acid substitution of S206R in *bla*_VIM-4_ and Q60R in *bla*_VIM-75_.

### 3.4. Plasmids

Three different plasmids were identified. One plasmid was present in all three isolates, another was present in two, and a unique plasmid was present only in one isolate (Figure 1).

Two closed plasmids, p401-1 (40,917 bps) and p401-2 (101,867 bps), were identified in the genome of Survcare401. Like Survcare401, Survcare357 harbored the identical 41 kb plasmid 1, p357-1, while Survcare372 harbored an additional third plasmid, p372-3 (46,639 bps), compared to Survcare401 (Figure 1). All plasmids of the isolates were non-typeable according to the plasmid Inc-group scheme, but a disrupted IncQ1 replicon CDS was detected on their chromosomes.

The 41 kb plasmids, e.g., p401-1, p357-1, and p372-1, encode the beta-lactamase *bla*_TEM-2_ gene located within a 5 kb Tn*801* transposon. A type IV secretion system, colicin immunity protein Cui, and type II toxin–antitoxin system (RelE/ParE) were identified (Figure 1). Plasmids with high sequence identity (>99.9%) and 100% coverage in *Proteus* and *Providence* strains have been frequently reported from different countries, such as p52260_1, p52808_2, P1-FZP3115, and p16Pre36_1 from Australia, China, and the Czech Republic (GenBank accession numbers: CP070570, CP070574, CP098451, and KX832926). Notably, they were only isolated after 2019. As p-1 is present in all three isolates studied, it appears to be commonly conserved in clinical *P. mirabilis* isolates.

The plasmids p401-2 and p372-2 both encoded *lnu(F)*, *aadA1*, and *aph(3′)-Ia*, conferring antibiotic resistance to lincomycin and aminoglycosides like streptomycin, kanamycin as well neomycin, respectively. In comparison, *aph(3′)-Ia* was embedded within a putative composite transposon, cn_2938_IS*26*. A type IV secretion system was also found in these plasmids. We found several plasmids with high identities (>99%) to the backbone of our p_2 plasmids, such as pIB-NDM-1, pPp47, pPv1-49741, and p06-1619-1 and p-dmpro_5749a_NDM1 from *Proteus* strains in Italy, Australia, the USA, and Bangladesh, respectively (GenBank accession numbers: CP045540, MG516912, CP104122, KX832929, and CP095677).

The third plasmid of Survcare372, p372-3, that harbored the class 1 integron encoding both *bla*_VIM-75_ and *bla*_VIM-4_ and other ARGs exhibited a backbone structure similar to plasmid sequences from different *Enterobacterales* species like *E. coli*, *Enterobacter* spp., *Klebsiella* spp., and *Providencia* spp. Interestingly, none of them carried the *bla*_VIM-75_ and *bla*_VIM-4_ encoding class 1 integron.

### 3.5. Virulence Factor Genes

Forty-six virulence genes were predicted in the genome of Survcare401 and forty-four genes each were predicted for Survcare372 and Survcare357 on their chromosomes (Appendix A). They comprise a type VI secretion system, flagella biosynthesis, fimbria syntheses, fimbria adhesin, metalloproteases, iron transporter, and Fe-S cluster assembly. The BRIG genome comparison (Figure 2) shows the different virulence groups on the chromosomes of the isolates. In the chromosome of Survcare357, the region coding a prophage was not detected but was present in the genome of Survcare372 and Survcare401.

Notably, a type I-E CRISPR-Cas system was present in all isolates, including *Cas3*, *CasABCDE*, *Cas1e*, *Cas2e*, and a CRISPR array consisting of 14 repeats, each 29 nucleotides in length, with a spacer length of 32 nucleotides, which typically provides *P. mirabilis* with immunity against bacteriophages.

### 3.6. Phylogeny

The phylogenetic analysis revealed that the isolates are phylogenetically not related to each other, indicating different origins (Figure 3). Nevertheless, the best-matching type-strain was the genome of *P. mirabilis* strain ATCC 29906 with an Average Nucleotide Identity (ANI) of 99.31% to 99.33% and a coverage of 0.86, which was identical for all three isolates.

## 4. Discussion

The co-occurrence of carbapenemase genes *bla*_VIM-4_ and *bla*_VIM-75_ in *P. mirabilis* has never been reported before. Herein, we have for the first time genomically characterized three *P. mirabilis* isolates co-harboring these two genes. Whole-genome analysis revealed that these genes were located on a class I integron in all three cases, which itself was located chromosomally in two isolates and plasmids in the third isolate. The three isolates originated from independent sources and are not genetically clonally related. We therefore suggest that the class I integron with the carbapenemase genes was acquired independently from different origins by the *P. mirabilis* strains.

Verona integron-encoded metallo-β-lactamases (VIM) in general are probably better known for *Pseudomonas aeruginosa* but have also been found in *Enterobacterales*, including *Proteus*, as well [6,22,23,24,25]. Currently, about 80 different VIM types have been identified, and the most reported genes are *bla*_VIM-1_ and *bla*_VIM-4_. *P. mirabilis* has been described as carrying *bla*_VIM-1_ in Greece, Bulgaria, and the Netherlands, for example [8,26,27]. The presence of the *bla*_VIM-4_ gene (a single amino acid mutation of the *bla*_VIM-1_ gene (S206R)) has also been described in *P. mirabilis*, for example, in Greece [28]. The *bla*_VIM-75_ gene is also a single amino acid mutation of *bla*_VIM-1_ (Q60R) but, to the best of our knowledge, has not been described in this species before.

We identified potential sources or potentially similar, parallel, acquisitions of *bla*_VIM-75_ by *P. mirabilis* in data in public databases: *bla*_VIM-75_ has apparently been detected in a *P. mirabilis* strain 07C16CRGN002 from a surveillance project in Canada (access no. NG_076844.1). Additionally, entries with an identical nucleotide sequence were found in an isolate from urine in Poland (designated as In2239, access no. OQ116828.1) and an animal isolate (*Gallus*) from Iraq (access no. LC848470.1).

The three isolates we studied herein harbored both *bla*_VIM-4_ and *bla*_VIM-75_, so the question arises as to how this double-carbapenemase status came to pass or how it was transmitted. We found a similar genetic construct in the literature, within a *Vibrio cholerae* strain described by Aberkane et al. [29]: a strain isolated from Yellow-legged gulls (*Larus michaellis)* co-carried *bla*_VIM-1_ and *bla*_VIM-4_. They were located on a class I integron with *aac(6′)-IIc* in between and a *sul1*-gene downstream, carried by its IncA/C plasmid. However, apart from this similarity, we have no further data to conclude that our isolates have an environmental epidemiological link.

Even though the specific combination of genes detected herein has never been published before, the genetic vicinity has been described in *P. mirabilis*; the genes are located on a class I integron, which is common in *P. mirabilis* carrying *bla*_VIM-1_. These integrons can carry other resistance genes as well, like *aacA7*, *dhfr*, and *aadA* [7]. Our isolates each carry an almost identical class I integron with the gene cassettes [*bla*_VIM-75_—*aac(6′)-IIc*—*bla*_VIM-4_—∆*qacE*—*sul1*], downstream to the integrase (Figure 1). We speculate that the three isolates may independently have acquired the same integron from an unknown source and that this integron is a development from previously known types of *P. mirabilis* integrons, through gene replacement, and possibly duplication with a mutation, as in the case of *bla*_VIM-4_ and *bla*_VIM-75_.

IS26 was not found in the integron structure, so high-level carbapenem resistance from the increased expression of the carbapenemase gene VIM through an increase in its copy number via the association of IS26 with the integron was not to be expected, as demonstrated in a previous study on *bla*_VIM-1_-carrying *P. mirabilis* [30].

Our isolates also harbored a large number of acquired ARGs which conferred resistance to several antibiotic classes. Acquired fluoroquinolone resistance genes were not found (Table 2), but amino acid substitutions in DNA gyrase subunit A GyrA (S83I) and topoisomerase IV ParC (S84I), compared to *P. mirabilis* HI4320 (NC_010554), were identified. Such alterations result in decreased susceptibility to fluoroquinolones in *Enterobacterales* [31,32], including *P. mirabilis* [33,34].

As far as the pathogenicity of the strains themselves is concerned, we have only limited clinical data. One was isolated from a rectal swab and another from a groin swab, which leaves little room for speculation. However, one isolate originates from a blood culture, which suggests the pathogenic potential of Survcare372. This is in accordance with the results of our virulence gene analysis, which revealed (for all three isolates) a plethora of common genes associated with *P. mirabilis* pathogenicity such as fimbriae, flagella, and gene clusters of type III, IV, and VI secretion systems. An interesting feature of our isolates is a type I-E CAS system in their genomes. CRISPRCas systems have been found only in about one-third of sequenced *P. mirabilis* genomes [1].

Plasmid type 1 (41 kb) in all three of our isolates seems to be a conserved common genome component of *P. mirabilis*, as we found many matching entries from *P. mirabilis in* the NCBI genome database. Plasmid type 2 (101 kb), which is present in two isolates, is also similar to numerous plasmids in the species, for example, pPp47 from silver gulls [35]. Notably, the segment containing the *tra*-operon is frequently found in strains of both human and environmental origin.

The overall increase in carbapenemases found in *Enterobacterales* constitutes a potential threat to modern healthcare facilities. National surveillance programs seek to quantify and assess the extent of this spread. Studies have shown that the spread of multidrug resistance, including carbapenem resistance, is associated with multiple species with different genetic backgrounds. Surveillance programs serve to identify early emerging resistance patterns and the emergence and spread of carbapenemase genes. Herein, we report rather unusual findings from a surveillance program based on WGS in Hesse, Germany: we identified three *P. mirabilis* isolates, each possessing a unique genetic element that harbors both *bla*_VIM-75_ and *bla*_VIM-4_ genes.

From a clinical perspective, our findings demonstrate a worrisome development: carbapenemase-producing *P. mirabilis* was previously rare in Germany. However, the detection of three strains with a combination of two carbapenemase genes could be a potential entry point for the spread of carbapenem resistance in *P. mirabilis* in clinical care in larger numbers.

Notably, the integron with these genes integrated into the chromosome in two of the three isolates and into a plasmid in one isolate. This indicated that the VIM-metallo-β-lactamase could easily spread by the integron both vertically and horizontally as they all carried the transferable conjugative plasmids (plasmid type 1 and type 2).

The present study has limitations. The first of which is that we only collected clinical strains from human medicine samples voluntarily submitted by participating laboratories. Thus, we can be wary of a potential threat in clinical care, but we have limited data about the full extent of the problem. It would be interesting to know if strains like ours can be found in the general population as well. Likewise, it is possible that additional data from animal or environmental sources could elucidate the origin of these strains and/or of the integron.

In summary, our results reveal dual carbapenemase-carrying *P. mirabilis* strains (*bla*_VIM-4_ and *bla*_VIM-75_) in Hesse, Germany, which may disseminate and complicate patient treatment by conferring carbapenem resistance. A mobile integron facilitates this resistance and may allow further dissemination to various *Enterobacterales*, particularly those of greater clinical relevance, such as *Escherichia coli* and *Klebsiella pneumoniae*. Furthermore, our data could support global efforts to trace the acquisition of two carbapenemase genes, *bla*_VIM-75_ and *bla*_VIM-4_ in the simultaneous presence of the 16S-rRNA-methylase gene *armA* by this species.

## 5. Conclusions

Herein, we report for the first time the complete genomes and the chromosomal and plasmidic co-occurrence of *bla*_VIM-4_ and *bla*_VIM-75_ of three multidrug-resistant and highly virulent *P. mirabilis* isolates from patients in Germany. The comparable genetic organization of the VIM gene-carrying class I integron structure with a previously documented variant (with *bla*_VIM-1_—and *bla*_VIM-4_) in a *V. cholerae* strain from wild avians suggests the potential for zoonotic transmission of VIM-carrying organisms, which may contribute to the spread of carbapenem resistance in humans. The high incidence of VIM-carrying *Proteus* in the environment underscores the significance of a One Health perspective in controlling multidrug resistance in *Enterobacterales*.

## Figures and Tables

**Figure 1 microorganisms-13-00266-f001:**
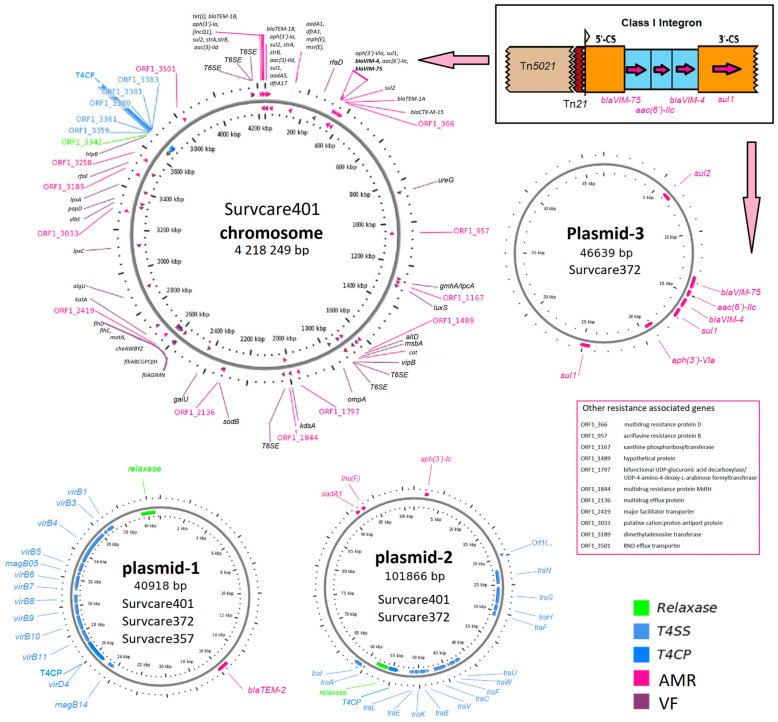
Genetic maps of the closed chromosome of Survcare401 and the plasmids of the isolates Survcare401, Survcare357, and Survcare372 show the antimicrobial resistance genes (ARGs) and virulence factor predicted as well as the Type 4 secretion system (T4SS), with the genetic structure of the class 1 integron encoding both *bla*_VIM-75_ and *bla*_VIM-4_ (top right). The arrows indicate the position of the class 1 integron on the chromosome (to the left) and the plasmid (downwards). Plasmid-1 represents plasmids p410-1, p357-1, and p372-1 presented in three strains, plasmid-2 represents p401-2 and p372-2 in strains Survcare402 and Survcare372, and plasmid-3 indicates p372-3, which was exclusively detected in Survcare372.

**Figure 2 microorganisms-13-00266-f002:**
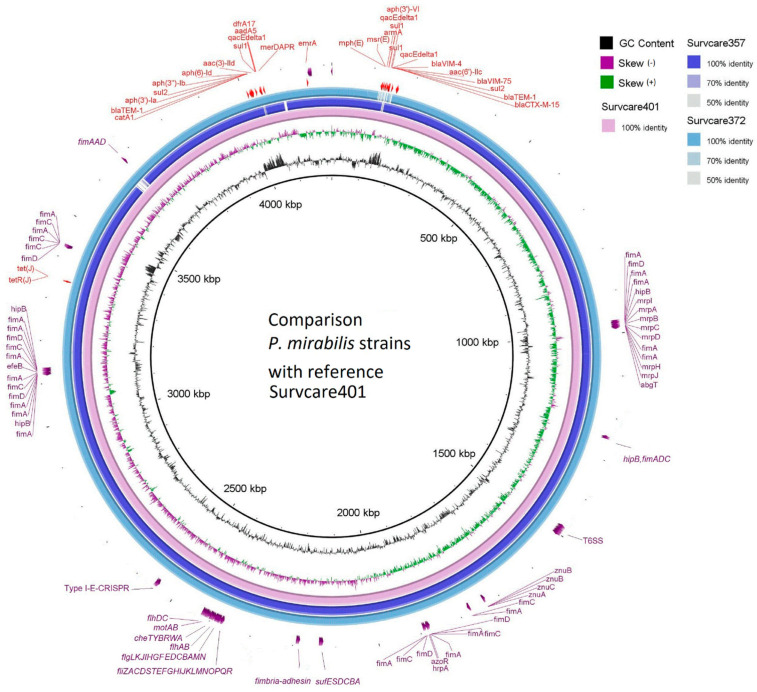
Comparison of the chromosomes of three isolates with Survcare401 as the reference sequence, indicating the presence of different virulence groups. The *bla*_VIM-75_- and *bla*_VIM-4_-carrying class 1 integron was present in the chromosomes of Survcare401 and Survcare357, and it was also present in Survcare372 on plasmid p372-3.

**Figure 3 microorganisms-13-00266-f003:**
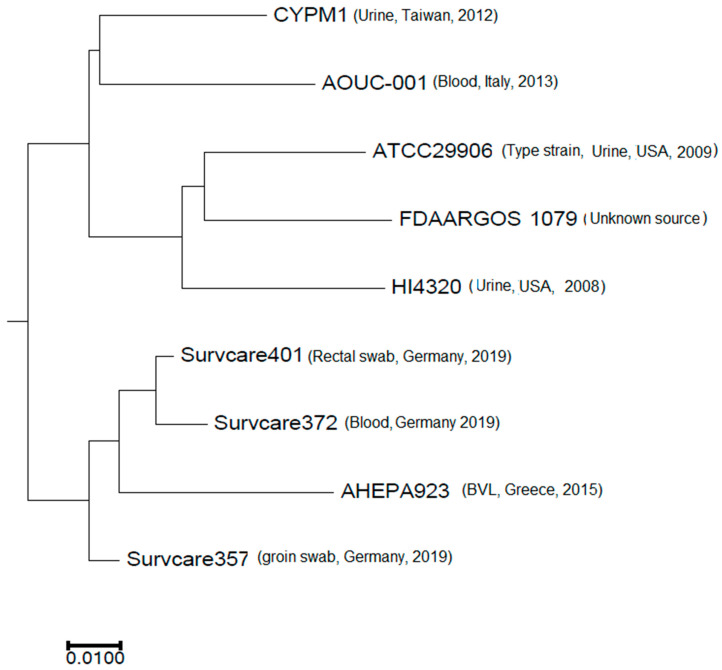
Phylogenetic comparison of the three isolates studied with selected publicly available complete *Proteus mirabilis* genomes.

**Table 1 microorganisms-13-00266-t001:** Sources and antibiotic susceptibility testing results of the *P. mirabilis* isolates. VITEK^®^2 MIC (mg/L) (interpretation).

Isolates	Survcare357	Survcare372	Survcare401
Isolate source	Groin swab	Blood	Rectal swab
Patient ages (years)	82	80	91
Patient sex	Female	Male	Male
Imipenem	≥16 [R] (>32)	≥16 [R] (12)	8 [R] (6)
Meropenem	≥16 [R] (8)	4 [I] (0.1)	1 [I] (0.125)
Ertapenem	≥8 [R]	4 [I]	≤0.12 [S]
Tigecycline	R	R	ND
Trimethoprim + Sulfamethoxazol	≥320 [R]	≥320 [R]	ND
Ampicillin	≥32 [R]	≥32 [R]	≥32 [R]
Ampicillin + Sulbactam	≥32 [R]	≥32 [R]	≥32 [R]
Piperacillin	≥128 [R]	≥128 [R]	≥128 [R]
Piperacillin + Tazobactam	≥128 [R]	8 [R]	8 [R]
Cefepim	R	R	R
Cefpodoxim	R	R	R
Cefotaxime	≥64 [R]	≥64 [R]	16 [R]
Ceftazidime	≥64 [R]	≥64 [R]	≥64 [R]
Cefuroxim	≥64 [R]	≥64 [R]	≥64 [R]
Aztreonam	R	R	R
Ciprofloxaxin	≥4 [R]	≥4 [R]	0.5 [I]
Moxifloxacin	R	R	R
Ofloxacin	R	R	R
Gentamicin	R	≥16 [R]	≥16 [R]

MIC, minimum inhibitory concentration; S, susceptible; I, susceptible with increased exposure; R, resistant. Numbers in brackets indicate MIC results from E-tests.

**Table 2 microorganisms-13-00266-t002:** Genomic characteristics, genome sizes, and antimicrobial resistance genes (ARGs).

Isolate	Genomic Features	Assembly Status	Size (bp)	ARGs *
Survcare401	Chromosome	circular	4,218,249	**β–lactams:** *bla*_VIM-75_, *bla*_VIM-4_, *bla*_CTX-M-15_, *bla*_TEM-1A_, *bla*_TEM-1B_ **Aminoglycosides:** *aadA1*, *aadA5*, *armA*, *aac(6′)-IIc*, *aac(3)-IId*, *aph(3′)-Ia*, *aph(3′)-VI*, *strA*, *strB* **Amphenicol:** *cat*, *catA1* **Sulphonamide:** *sul1* (3x), *sul2* (2x) **Tetracycline:** *tet(J)* **Macrolide/Streptogramin B:** *msr(E)*, *mph(E)* **Trimethoprim:** *dfrA1*, *dfrA17* **Quaternary ammonium compound:** *qac*ΔE (3x)
	Plasmid p401-1	circular	40,917	**β–lactams:** *bla*_TEM-2_
	Plasmid p401-2	circular	101,867	**Aminoglycosides:** *aadA1*, *aph(3′)-Ic* **Lincosamide:** *lnu(F)*
Survcare357	Chromosome	linear	~4,191,000	**β–lactams:** *bla*_VIM-75_, *bla*_VIM-4_, *bla*_CTX-M-15_, *bla*_TEM-1A_, *bla*_TEM-1B_ **Aminoglycosides:** *aadA1*, *armA*, *aac(6′)-IIc*, *aac(3)-IId*, *aph(3′)-Ia*, *aph(3′)-VI*, *strA*, *strB* **Amphenicol:** *cat*, *catA1*, **Sulphonamide:** *sul1*, *sul2* (2x) **Tetracycline:** *tet(J)* **Macrolide/Streptogramin B:** *msr(E)*, *mph(E)* **Trimethoprim:** *dfrA1* **Quaternary ammonium compound:** *qac*ΔE (3x).
	Plasmid p357-1	circular	40,920	**β–lactams:** *bla*_TEM-2_
Survcare372	Chromosome	linear	~4,165,000	**β—lactams:** *bla*_CTX-M-15_ (2x), *bla*_TEM-1A_, *bla*_TEM-1B_ **Aminoglycosides:** *aadA1*, *aadA5*, *aac(3)-IId*, *aph(3′)-Ia*, *strA*, *strB* **Amphenicol:** *cat*, *catA1* **Sulphonamide:** *sul1* **Tetracycline:** *tet*(J) **Trimethoprim:** *dfrA1*, *dfrA*17 **Quaternary ammonium compound:** *qac*ΔE.
	Plasmid p372-1	circular	40,918	**β–lactams:** *bla*_TEM-2_
	Plasmid p372-2	circular	101,866	**Aminoglycosides:** *aadA1*, *aph(3′)-Ic* **Lincosamide:** *lnu(F)*
	Plasmid p372-3	circular	46,639	**β—lactams:** *bla*_VIM-75_, *bla*_VIM-4_ **Aminoglycosides:** *armA*, *aac(6′)-IIc*, *aph(3′)*-VIa **Sulphonamide:** *sul1 (2x)*, *sul2* **Macrolide/Streptogramin B:** *msr(E)*, *mph(E)* **Quaternary ammonium compound:** *qac*ΔE (2x).

* The bold terms Aminoglycosides, Amphenicol, β—lactams, Macrolide/Lincosamide/Streptogramin B, Sulphonamide, Trimethoprim and Tetracycline represent the different classes of antibiotic, to which ARGs confer resistance. The quaternary ammonium compound is a disinfectant.

## Data Availability

The complete genome sequences of the *P. mirabilis* isolates have been deposited at the National Center for Biotechnology Information (NCBI) under accession no. JAFHGA01 for Survcare357, JAFHGO01 for Survcare372, and CP177077-CP177079 for Survcare401, within the Bioproject accession no. PRJNA692829.

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
