# Peer review of "The Detection of Extensively Drug-Resistant Proteus mirabilis Strains Harboring Both VIM-4 and VIM-75 Metallo-β-Lactamases from Patients in Germany"

_microorganisms, 2025, doi:10.3390/microorganisms13020266_

Round 1

Reviewer 1 Report

Comments and Suggestions for Authors

Thank you for the opportunity to review the manuscript titled “Detection of Multidrug-Resistant Proteus mirabilis Strains Harboring Both VIM-4 and VIM-75 Metallo-β-lactamases from Patients in Germany.” The study addresses a critical and timely issue by characterizing multidrug-resistant P. mirabilis strains and highlighting the genetic mechanisms underlying carbapenem resistance. The findings have significant implications for public health and antimicrobial resistance surveillance.

While the study provides valuable insights, significant revisions are needed to improve the manuscript’s organization, clarity, and accuracy. Addressing these issues will enhance the study's scientific rigor and readability.

Abstract

1.      The abstract provides a concise summary of the study, but it could benefit from improved organization. While the background, methods, and findings are addressed, the abstract would be clearer if it explicitly outlined the problem, methods, key results, and implications in a structured order.

2.      Several sentences were too long and need to be conscious and short for more clarity

3.      In the results section, consider explaining briefly why these particular three isolates were chosen for sequencing.

4.      The conclusion should be restructured to a more conscious and readable form.

 Introduction

1.      The introduction provides a good overview of the clinical relevance of Proteus mirabilis and its resistance mechanisms. However, the transition between topics should be smoother.

2.      Avoid redundancy and repeated information.

3.      In line 70, the authors said, " We carried out a surveillance study of multi-resistant Gram-negative bacteria in the state of Hesse in Germany, in which carbapenem-resistant Enterobacterales collected within a three-year period were subjected to whole genome sequencing [10]". Please avoid including citations in sentences that describe the impact of the current study.

4.      The text switches between abbreviations and full terms of bacteria (multidrug-resistant vs MDR). Ensure consistent usage throughout.

5.       Several sentences like "Proteus are members of the family Morganellaceae that exhibit some level of intrinsic resistance to imipenem – however, this can usually be overcome with high dosage" are too long and complex. Please split these sentences for clarity.

Material and Methods

1.      The methods section is detailed, but the information could be better organized into clear subsections (e.g., "Study Design," "Antimicrobial Susceptibility Testing," "Whole Genome Sequencing," etc.).

2.      The manuscript does not provide an IRB approval code or reference number, which is critical for ensuring compliance with ethical research standards. This must be addressed to confirm that the study adheres to necessary ethical guidelines. The authors should either include this information or clarify why it is not applicable.

3.      Please clarify the criteria used to classify strains as multidrug-resistant.

4.      I noticed a discrepancy between the number of Proteus mirabilis isolates mentioned in the abstract and the materials and methods section. In the materials and methods, it is stated that five P. mirabilis isolates were obtained during the surveillance study, whereas the abstract mentions the sequencing of three P. mirabilis isolates. This inconsistency could lead to confusion regarding the scope of the analysis.

Results

1.      The results are comprehensive but sometimes overwhelming due to excessive technical jargon. Simplify language for clarity.

2.      Tables and figures are useful, but ensure they are self-explanatory. For example, Table 1 could benefit from a more descriptive caption.

3.      Some sentences need to be clearer, for example, "The colistin test indicate a MIC of >64 µg/ml" should be "The colistin test indicates an MIC of >64 µg/ml."

Discussion

1.      The discussion effectively connects the findings to the broader context of antimicrobial resistance. However, it could benefit from a more structured approach:

·         Start by summarizing the key findings.

·         Discuss the implications for public health and clinical practice.

·         At the end of discussion, highlight limitations and future directions.

2.      Sentences like "Our three isolates do not appear to be clonally related, suggesting independent acquisition of these resistance determinants" could be rephrased for conciseness.

Conclusion

3.      Emphasize that this is the first report of complete genome sequences of P. mirabilis isolates harboring both blaVIM-4 and blaVIM-75 to highlight the novelty of the findings.

4.      Provide a clearer connection between the comparison to Vibrio cholerae and the hypothesis of zoonotic transmission to strengthen this point.

5.      Strengthen the message about the importance of a one-health approach by more explicitly linking it to the broader implications for both human and animal health.

Minor comments

The manuscript contains several overly complex sentences and unclear phrasing, which affects readability.

Author Response

Comments and Suggestions for Authors

Thank you for the opportunity to review the manuscript titled “Detection of Multidrug-Resistant Proteus mirabilis Strains Harboring Both VIM-4 and VIM-75 Metallo-β-lactamases from Patients in Germany.” The study addresses a critical and timely issue by characterizing multidrug-resistant P. mirabilis strains and highlighting the genetic mechanisms underlying carbapenem resistance. The findings have significant implications for public health and antimicrobial resistance surveillance.

While the study provides valuable insights, significant revisions are needed to improve the manuscript’s organization, clarity, and accuracy. Addressing these issues will enhance the study's scientific rigor and readability.

We thank the reviewer for his time and comments to improve the manuscript.

  1. Abstract
    • The abstract provides a concise summary of the study, but it could benefit from improved organization. While the background, methods, and findings are addressed, the abstract would be clearer if it explicitly outlined the problem, methods, key results, and implications in a structured order.

Responses of authors: Many thanks for these suggestions. In principle, the structure of the summary already corresponds to the requested form. For a better readability we will follow the recommendation of the other reviewer (Reviewer 2) to omit the subtitles like “Background”, “Methods”, “Results” and “Conclusion”.

1.2. Several sentences were too long and need to be conscious and short for more clarity

1.3. In the results section, consider explaining briefly why these particular three isolates were chosen for sequencing.

1.4. The conclusion should be restructured to a more conscious and readable form.

Responses of authors to 1.2. to 1.4: Amendment followed. We have rephrased some of these sentences. (see Lines 19-22, 25-27, 29-30 in the revised manuscript)

  1. Introduction

2.1. The introduction provides a good overview of the clinical relevance of Proteus mirabilis and its resistance mechanisms. However, the transition between topics should be smoother.

Responses of authors: Many thanks for these suggestions. These suggestions have been accepted and considered (see, lines 49, 59 and 66).

2.2. Avoid redundancy and repeated information.

Responses of authors: Many thanks for these suggestions. It has been checked and to removed redundancy and repeated information.

2.3. In line 70, the authors said, " We carried out a surveillance study of multi-resistant Gram-negative bacteria in the state of Hesse in Germany, in which carbapenem-resistant Enterobacterales collected within a three-year period were subjected to whole genome sequencing [10]". Please avoid including citations in sentences that describe the impact of the current study.

Responses of authors: Many thanks for these suggestions. The citation was removed (see line 68 in the revised manuscript).

2.4. The text switches between abbreviations and full terms of bacteria (multidrug-resistant vs MDR). Ensure consistent usage throughout.

Responses of authors: Many thanks for these suggestions. These suggestions have been accepted. The MDR abbreviations have been removed (see line 47).

2.5. Several sentences like "Proteus are members of the family Morganellaceae that exhibit some level of intrinsic resistance to imipenem – however, this can usually be overcome with high dosage" are too long and complex. Please split these sentences for clarity.

Responses of authors: Many thanks for these suggestions. These suggestions have been accepted and inserted.

  1. Material and Methods

3.1. The methods section is detailed, but the information could be better organized into clear subsections (e.g., "Study Design," "Antimicrobial Susceptibility Testing," "Whole Genome Sequencing," etc.).

Responses of authors: Many thanks for these suggestions. These suggestions have been accepted and inserted. The titles for the subsections have been added (Lines 74, 86, 96 and 111). 

3.2. The manuscript does not provide an IRB approval code or reference number, which is critical for ensuring compliance with ethical research standards. This must be addressed to confirm that the study adheres to necessary ethical guidelines. The authors should either include this information or clarify why it is not applicable.

Responses of authors: Many thanks for these suggestions. These suggestions have been accepted and inserted. We have added an “Ethical approval” section (lines 111-115) with the corresponding note.

3.3. Please clarify the criteria used to classify strains as multidrug-resistant.

Responses of authors: Many thanks for these suggestions. These suggestions have been accepted and the criteria for the classification of MDR, XDR and PDR have been added (see lines 92-95).  Our three isolates analyzed were ‘non-susceptible to at least 1 agent in all but 2 or fewer antimicrobial categories’ and therefore belong to the XDR (Extensively Drug-Resistant). We modified the manuscript title accordingly.

3.4. I noticed a discrepancy between the number of Proteus mirabilis isolates mentioned in the abstract and the materials and methods section. In the materials and methods, it is stated that five P. mirabilis isolates were obtained during the surveillance study, whereas the abstract mentions the sequencing of three P. mirabilis isolates. This inconsistency could lead to confusion regarding the scope of the analysis.

Authors´replies: Many thanks for this question. As part of the surveillance study, five phenotypically carbapenem-resistant P. mirabilis isolates were recovered and undergone whole-genome-sequencing using short-read sequencing technology. Three isolates with duplicated VIM-alleles were identified during genome analysis, and no known carbapenemase genes were found in two other isolates. Long-read sequencing was then performed for the three VIM-positive isolates to analyze the genetic positions of the VIM-genes, etc., which was the subject of the present study. A corresponding explanation has been added to the text (see lines 77-82)

  1. Results

4.1. The results are comprehensive but sometimes overwhelming due to excessive technical jargon. Simplify language for clarity.

Autor´s responses: Many thanks for these suggestions. The suggestions have been accepted and inserted. Amendments, for example, in lines 176-179 and 185-190.

4.2. Tables and figures are useful, but ensure they are self-explanatory. For example, Table 1 could benefit from a more descriptive caption.

Autor´s responses: Many thanks for these suggestions. The heading and footnote of Table 1 have been extensively revised, including the dimensions of the values (see Lines 142 to 145).

4.3. Some sentences need to be clearer, for example, "The colistin test indicate a MIC of >64 µg/ml" should be "The colistin test indicates an MIC of >64 µg/ml."

Autor´s responses: Many thanks for these suggestions. The suggestions have been accepted and inserted.

  1. Discussion

5.1. The discussion effectively connects the findings to the broader context of antimicrobial resistance. However, it could benefit from a more structured approach:

 Start by summarizing the key findings.

  • Discuss the implications for public health and clinical practice.
  • At the end of discussion, highlight limitations and future directions.

Autor´s responses: Many thanks for these suggestions. The suggestions have been accepted and incorporated.   

The discussion now opens with the main findings (lines 224- 288), followed by the implications for public health and clinical practice (289-306), and ends by pointing out the limitations of the study. (307-312).

5.2. Sentences like "Our three isolates do not appear to be clonally related, suggesting independent acquisition of these resistance determinants" could be rephrased for conciseness.

Autor´s responses: Many thanks for these suggestions. The suggestions have been accepted and inserted. This sentence has been changed to “Our three isolates were not clonally related, suggesting independent acquisition of these resistance determinants"

Conclusion

5.3. Emphasize that this is the first report of complete genome sequences of P. mirabilis isolates harboring both blaVIM-4 and blaVIM-75 to highlight the novelty of the findings.

5.4. Provide a clearer connection between the comparison to Vibrio cholerae and the hypothesis of zoonotic transmission to strengthen this point.

5.5. Strengthen the message about the importance of a one-health approach by more explicitly linking it to the broader implications for both human and animal health.

Autor´s responses to 5.3-5.5: Thank you for these suggestions. The suggestions have been accepted and added. The “conclusion” section has been amended (see lines 322-324, and 328-329).

Minor comments

The manuscript contains several overly complex sentences and unclear phrasing, which affects readability.

Autor´s responses: Many thanks for these suggestions. These have been accepted and added.

Reviewer 2 Report

Comments and Suggestions for Authors

On request of Microorganisms, I have revised the manuscript titled “Detection of Multidrug-Resistant Proteus mirabilis Strains Harboring Both VIM-4 and VIM-75 Metallo-β-lactamases from Patients in Germany”, by Moritz Fritzenwanker and colleagues.

With the aim at limiting the trasmission of carbapenem resistance in P. mirabilis, nowaday resistant to almost all available antibiotics including colistin, the Authors, for the first time, sequenced the genomes of three carbapenem-resistant P. mirabilis isolates obtained during a surveillance study in Germany. To this end, they used both short-read and long-read sequencing techniques, followed by detailed genomic analyses, finding that they commonly harbored both blaVIM-4 and blaVIM-75 genes on a class I integron, located in two cases on the chromosome and in one case on a plasmid. Such results emphasized the key role of class 1 integrons in the transmission of carbapenem resistance in P. mirabilis.

COMMENTS

Althought carbapenem resistance is generally rare in Proteus mirabilis, it is increasingly increasing and cases of presence of genes encoding for carbapenem resistance have been recorded in several countries, thus becoming a problem of worldwide concern. Therefore, considering that Proteus mirabilis is a well-known opportunistic pathogen predominantly associated with urinary tract infections (UTIs) nowadays exhibiting resistance to multiple antibiotics, including last-resort options like colistin, works investigating the genomes of carbapenem-resistant P. mirabilis isolates to find which genes are present in all isolates, and where are located, as possible responsible of resistance trasmission, could be highly relevant for the development of new targeted antibiotics finalized to limit the spread of resistance.

The topic is interesting, and the study design appears rational and well described. Anyway, some minor concerns, force me to not accept this paper in the current form and to ask for minor revisions. Please, consider the following list.

Abstract. Please, remove the titles of sections such as Background, Methods etc.

Please, check if for all reagent and instruments, it has been reported both the name of producer and its location, including city and country.

Please, remove useless bold for words such as Table and Figure in all manuscript. As an example, Table 1 in lines 108-109.

Table 1. Please, insert a colon in place of the dot after the "S" in the footnotes. Add also a specification for the number values present in the rows of antibiotics and their measure unit.

Conclusions need improvement to better emphasize the discoveries.

The reference list needs to be corrected in the format according to the template provided by microorganisms.

Author Response

Comments and Suggestions for Authors

On request of Microorganisms, I have revised the manuscript titled “Detection of Multidrug-Resistant Proteus mirabilis Strains Harboring Both VIM-4 and VIM-75 Metallo-β-lactamases from Patients in Germany”, by Moritz Fritzenwanker and colleagues.

With the aim at limiting the trasmission of carbapenem resistance in P. mirabilis, nowaday resistant to almost all available antibiotics including colistin, the Authors, for the first time, sequenced the genomes of three carbapenem-resistant P. mirabilis isolates obtained during a surveillance study in Germany. To this end, they used both short-read and long-read sequencing techniques, followed by detailed genomic analyses, finding that they commonly harbored both blaVIM-4 and blaVIM-75 genes on a class I integron, located in two cases on the chromosome and in one case on a plasmid. Such results emphasized the key role of class 1 integrons in the transmission of carbapenem resistance in P. mirabilis.

COMMENTS

Althought carbapenem resistance is generally rare in Proteus mirabilis, it is increasingly increasing and cases of presence of genes encoding for carbapenem resistance have been recorded in several countries, thus becoming a problem of worldwide concern. Therefore, considering that Proteus mirabilis is a well-known opportunistic pathogen predominantly associated with urinary tract infections (UTIs) nowadays exhibiting resistance to multiple antibiotics, including last-resort options like colistin, works investigating the genomes of carbapenem-resistant P. mirabilis isolates to find which genes are present in all isolates, and where are located, as possible responsible of resistance trasmission, could be highly relevant for the development of new targeted antibiotics finalized to limit the spread of resistance.

The topic is interesting, and the study design appears rational and well described. Anyway, some minor concerns, force me to not accept this paper in the current form and to ask for minor revisions. Please, consider the following list.

Autor´s responses: We thank the reviewer for his time and comments to improve the manuscript.

Abstract. Please, remove the titles of sections such as Background, Methods etc.

Autor´s responses: Many thanks for these suggestions. The titles of the sections have been removed. See also Reviewer 1, numbers 1.1. to 1.4.

Please, check if for all reagent and instruments, it has been reported both the name of producer and its location, including city and country.

Autor´s responses: Many thanks for these suggestions. These suggestions have been accepted and inserted.

Please, remove useless bold for words such as Table and Figure in all manuscript. As an example, Table 1 in lines 108-109.

Autor´s responses: Many thanks for these suggestions. The suggestions have been accepted and inserted.

Table 1. Please, insert a colon in place of the dot after the "S" in the footnotes. Add also a specification for the number values present in the rows of antibiotics and their measure unit.

Autor´s responses: Many thanks for these suggestions. The suggestions have been accepted and inserted, see the responses to point 4.2. from reviewer 1.

Conclusions need improvement to better emphasize the discoveries.

Autor´s responses: Many thanks for these suggestions. The suggestions have been accepted and inserted. We have rephrased the conclusions section. (see lines 322-330)

The reference list needs to be corrected in the format according to the template provided by microorganisms.

Autor´s responses: Many thanks for these suggestions. These suggestions have been accepted and inserted.

Round 2

Reviewer 1 Report

Comments and Suggestions for Authors

Thanks to the authors.They adressed all comments.